# The Polysomnographical Meaning of Changed Sleep Quality—A Study of Treatment with Reduced Time in Bed

**DOI:** 10.3390/brainsci13101426

**Published:** 2023-10-07

**Authors:** Paolo d’Onofrio, Susanna Jernelöv, Ann Rosén, Kerstin Blom, Viktor Kaldo, Johanna Schwarz, Torbjörn Åkerstedt

**Affiliations:** 1Stress Research Institute, Department of Psychology, Stockholm University, 10691 Stockholm, Sweden; donofrio.paolo@yahoo.it (P.d.); johanna.schwarz@su.se (J.S.); 2Centre for Psychiatry Research, Department of Clinical Neuroscience, Karolinska Institutet, & Stockholm Health Care Services, Region Stockholm, 14186 Stockholm, Sweden; susanna.jernelov@ki.se (S.J.); ann.rosen@ki.se (A.R.); kerstin.blom@ki.se (K.B.); viktor.kaldo@ki.se (V.K.); 3Division of Psychology, Department of Clinical Neuroscience, Karolinska Institutet, 17177 Stockholm, Sweden; 4Department of Psychology, Faculty of Health and Life Sciences, Linnaeus University, 35195 Växjö, Sweden

**Keywords:** subjective sleep, objective sleep, ratings, PSG, sleep restriction, sleep compression

## Abstract

Background: Reports of poor sleep are widespread, but their link with objective sleep (polysomnography—PSG) is weak in cross-sectional studies. In contrast, the purpose of this study was to investigate the association between changes in subjective and objective sleep variables using data from a study of the reduction in time in bed (TIB). Methods: One sleep recording was carried out at baseline and one at treatment week 5 (end of treatment) (N = 34). Results: The Karolinska Sleep Quality Index improved and was correlated with improvement in sleep efficiency (r = 0.41, *p* < 0.05) and reduction in TIB (r = −0.47, *p* < 0.01) and sleep latency (r = 0.36, *p* < 0.05). The restorative sleep index showed similar results. Improvements in the insomnia severity index (ISI) essentially lacked correlations with changes in the PSG variables. It was suggested that the latter may be due to the ISI representing a week of subjective sleep experience, of which a single PSG night may not be representative. Conclusions: It was concluded that changes in the subjective ratings of sleep are relatively well associated with changes in the PSG-based sleep continuity variables when both describe the same sleep.

## 1. Introduction

Sleep problems are widespread in society, with around 10% in the general population suffering from insomnia, and more than twice as many reporting various indications of “disturbed sleep” [1]. Poor sleep is also linked to poor health, as is long or short sleep [1]. However, indicators of poor sleep are based on self-reports, and it is not clear as to what extent self-reported sleep problems reflect objective deviations from normal sleep, as measured with polysomnography (PSG). This question is of interest due to the links between sleep and health, and because of the more epistemological question of what people “mean” by poor (or good) sleep. However, studies comparing retrospective reports of longer periods of sleep with PSG variables are quite rare, and indicate very poor agreement between PSG and, for example, the Pittsburgh Sleep Quality Index (PSQI) [2] or the Uppsala Sleep Inventory [3]. An interesting observation in this context is that individuals’ perception of good or poor sleep referring to longer time periods seem less based on the perception of sleep continuity (difficulties falling asleep, repeated awakenings, amount of time awake, early morning awakenings, etc.), than on states like fatigue, feelings of being well rested, depression, or anxiety [4].

In contrast, studies of the association between PSG and sleep diary ratings in the morning after the recorded sleep usually present stronger associations. Thus, it appears that PSG sleep efficiency and other measures of sleep continuity (measures of the amount of sleep vs. awake during time in bed), but not sleep architecture (e.g., sleep stages), are associated with morning (“diary”) ratings of subjective sleep quality, describing good–poor sleep, or frequency of sleep problems [5,6,7,8,9,10,11]. The key rating in these studies was “sleep quality” (or similar), and it has been associated with PSG sleep efficiency or wake time after sleep onset (WASO—negatively), TST [7,8,11], awakenings [7,8,10], N1 % [7], and sleep stages N1 and N2 in minutes [6]. In two longitudinal studies, with experimentally displaced and shortened times in bed, the correlations over time were quite high for changes in rated sleep quality and changes in several PSG variables (including the total sleep time (TST) and stage N3) [12,13].

In the present study, we wanted to bring the longitudinal approach one step further by exploiting an experimental induction of (presumably) better sleep for insomnia patients, using sleep restriction or sleep compression, during a five-week treatment period [14] (see below). The specific purpose of the present study was to investigate the association between changes in subjective ratings of sleep and changes in standard PSG variables between the baseline week and the last treatment week. One focus was on changes in morning diary reports of sleep quality for the same sleep that was recorded polysomnographically. A second focus was on changes in PSG and insomnia severity ratings (ISI), the latter of which refer to a longer period of time [2,15], in the present case one week (containing the PSG recording). The two subgroups of the original study were combined into one group for the present purpose.

Based on previous work, we hypothesized that the association between changes in objective and subjective sleep from baseline to week 5 of treatment would be largest for PSG sleep continuity measures (sleep efficiency, WASO (overlapping with sleep efficiency), number of awakenings, TST, and sleep latency) and corresponding Karolinska Sleep Diary ratings, such as its Sleep Quality Index and its components “calm sleep”, “slept through”, “difficulties falling asleep”, and “sleep quality”. For the ISI, no specific hypothesis could be formulated, since prior work has not indicated any associations, but it was thought that the particular design (change of TIB) might bring out significant associations. This hypothesis was, thus, explorative.

## 2. Materials and Methods

The present study (N = 34) was part of a larger project (with 234 participants) aiming to compare two different treatments using reduced time in bed to improve sleep. A subset, living in the Stockholm area, was invited to participate with polysomnographic recordings of sleep (see below and the study published by the authors of [14]). This study was preregistered at clinicaltrials.gov (NCT02743338). It was approved by the Swedish Ethical Review Authority (Dnr 2016/44-31/4 and Dnr 2018/2025-32).

### 2.1. Participants and Recruitment

The recruitment for the entire study was open to adults with insomnia living in Sweden. Patients were mainly recruited through the homepage of the Internet Psychiatry Clinic, part of the public health system in Stockholm, Sweden. The criteria for inclusion were: ≥18 years, a score >10 on the insomnia severity index (ISI) [15], diagnosis of insomnia as per the criteria of the *Diagnostic and Statistical Manual of Mental Disorders, Fifth Edition* (DSM-5, American Psychiatric Association, 2013), the ability to read and write in Swedish, and the ability to respond to questionnaires online.

The exclusion criteria comprised the following: comorbid sleep disorders requiring treatment (sleep apnea or narcolepsy), bipolar disorder or other comorbid disorder contraindicative of sleep restriction, ongoing drug or alcohol abuse, medication with side effects on sleep (e.g., some anti-inflammatory medications), previous experience of sleep restriction therapy, sleep compression therapy, or similar methods, night-shift work, use of sleep medication in a way that could interfere with the treatment, and not completing the online pre-measurement on time. The use of antidepressants or sleep medication was not a reason for exclusion. Table 1 shows that the majority of the sample comprised women with moderately severe insomnia.

For the sub-study with PSG recordings, participants in the Stockholm area were asked if they were interested in participating. All who agreed to participate signed an informed consent form for this addition. A total of 36 agreed to participate, but due to a logistical error 2 sleep diaries were lost. Thus, 34 participants remained for analysis. The participants were described in detail in a previous paper [14]. However, Table 1 presents some of the key background data.

### 2.2. Procedure

This study started with several weeks of recruiting and screening procedures. This was followed by a baseline measurement, 5 weeks of therapist-supported treatment, and 5 weeks of unsupervised treatment (Figure 1). Sleep was recorded 5 times under the following time points: during habituation (before the start of the study), baseline week, and treatment weeks two, five, and ten (here, only results from the baseline week and week 5 were used). Each morning after the sleep recording, the Karolinska Sleep Diary was filled out. The exact position of the recording in the baseline or last treatment week varied depending on agreement with the patient, and exclusions of Friday, Saturday, and Sunday nights. The insomnia severity index was filled out at the end of the baseline week and at the end of each week of treatment. It is important to note that the number of days between the sleep recording and filling out the ISI could vary within the week studied.

Therapist support was received several times a week through written messages. Therapists were licensed psychologists or clinical psychology students under supervision with a licensed psychologist. The patients underwent either ‘sleep restriction therapy’ or ‘sleep compression therapy’. These methods have been described in detail elsewhere [14], but briefly, sleep restriction therapy involves an immediate restriction of TIB at the start of treatment. Sleep compression therapy refers to a gradual reduction across the treatment period. The goal of these two methods was to increase sleep efficiency, as calculated from sleep diary registrations. The goal was set to 85–90% sleep efficiency (based on self-reports). Patients evaluated their sleep efficiency weekly and adjusted their time in bed to approach a sleep efficiency of 90%. The therapist supported participants in both calculating the allowed time in bed and in adhering to the planned sleep times. Five weeks were expected to be enough for both methods to have an effect, and data could therefore be analyzed in a combined sample.

### 2.3. Subjective Ratings

The ISI was constructed to measure symptoms related to insomnia disorder and contains seven questions regarding the preceding seven days, three of which concern difficulties sleeping: “difficulties falling asleep”, “difficulties staying asleep”, and “waking up too early” (with response alternatives from “none” to “very severe” (0–4)) [2,15]. Four more questions concern “dissatisfaction with sleep”, “interference of sleep problems with daily functioning”, “how noticeable one’s sleep problems are by others”, and “how worried/distressed one is about one’s sleep problems”. The responses range from “very satisfied” to “very dissatisfied”, from “not at all interfering” to “very much interfering”, from “not at all noticeable” to “very much noticeable”, and from “not at all worried” to “very much worried” (all scored 0–4). Thus, the score of the ISI ranges from 0–28.

The Karolinska Sleep Diary (KSD) [12] contains the items “sleep quality” (“how did you sleep?”) with response alternatives (ranging from 1—very poorly to 5—very well), “ease falling asleep” (from 5—very easy to 1—very difficult), “calm sleep” (from 5—very calm to 1—very restless), and “slept throughout the time allotted” (from 5—yes to 1—not at all). For these variables, we also computed the mean score to construct the sleep quality index (SQI)**.** In addition, we used the items “ease of waking up” (ranging from 5—very easy to 1—very difficult), “sufficient sleep” (from 5—yes fully sufficient to 1—very insufficient), and “feeling refreshed after awakening” (from 5—completely to 1—not at all). For these variables, we computed a mean score to form a restorative sleep index (RSI). For both these indices, their scores range from 1 to 5, with a higher score being “better”. In addition, the KSD also included ratings of sleepiness, using the Karolinska Sleepiness Scale (KSS) [16] at bedtime and rising. The scale ranges from 1 (extremely alert) to 9 (indicating a very high level of sleepiness, fighting sleep, or exerting an effort to stay awake).

### 2.4. Polysomnography

All sleep recordings took place on working days. The first was used as “habituation”, recorded before baseline, to make the participant acquainted with the recording equipment and procedure. This recording was not included in the analyses. To rule out sleep apnea, all participants were screened for breathing pauses of ≥10 s with desaturations ≥3%, but no such participants were identified.

Standard sleep registration (according to the American Association of Sleep Medicine sleep scoring guidelines [17]) was performed with scalp EEG derivations placed on the central (C3 and C4) and frontal (F3 and F4) areas, referenced to the contralateral left (M1) or right (M2) mastoid process derivation, two electro-oculograms (EOGs) with oblique derivations placed on each outer canthus (1 cm below the left eye and 1 cm above the right eye), and one bipolar submental electromyographic (EMG) derivation. The electrode (Ag/AgCl electrode) montage and the impedance test, with a five kΩ maximum impedance, were carried out in the participants’ home by an experienced researcher, approximately 120 min before the usual bedtime. Sleep data were recorded on portable Embla recorders (Flaga HF®/Medcare) with a sampling rate of 256 Hz. The equipment was collected in the participants’ home in the morning following the sleep registration.

The sleep recordings were visually scored according to the guidelines of the American Academy of Sleep Medicine [17], using Embla® Remlogic™ software. Before sleep scoring, different filters were applied to the EEG, EOG, and EMG in accordance with the digital specifications of the AASM manual (Berry et al., 2012). The following standard PSG parameters were computed: time in bed (TIB), total sleep time (TST), minutes of sleep stages N1–3 and REM, the amount of time awake during sleep period time (WASO), that is, between sleep onset and final awakening, sleep efficiency (TST/TIB), number of awakenings and arousals (per hour of TST), number of sleep spindles (per hour of TST), sleep latency as time from ‘eyes closed’ to the first epoch of at least three consecutive sleep epochs (stage N1 or other sleep stages), time to first stage N3 from sleep onset (SWS latency), and time to first stage REM from sleep onset (REM latency). Arousals were scored using the American Sleep Disorders Association criteria [17,18]. An arousal from sleep was defined as an interruption of sleep stages N1–3 or REM for more than 3 s, and for less than 15 s. During REM sleep, an increase in EMG activity was required for scoring an arousal.

### 2.5. Statistical Analyses

Changes from baseline to treatment week 5 were assessed using a repeated measures analysis of variance (ANOVA), with time as the repeated measure. The main analysis consisted of correlations between changes (from baseline to treatment week 5) in PSG, the ISI, and the two sleep diary indices (the sleep quality Index and the restorative sleep index). To study the association on an item level between changes in the objective and subjective sleep variables, we correlated the change in PSG variables with change in the different sleep ratings. In addition, we analyzed a number of correlations that may be interesting for understanding the results. This included the correlations between the change and baseline values and between the baseline values and treatment week 5 values. It may be noted that in the text below we refer to statistical significance when we use the term “significance”. There are no available data on correlations between changes for the present type of variables, but using cross-sectional correlations as an estimate, for a power of 0.8. At a significance level of *p* < 0.05, for an assumed correlation of r = 0.50, we would need a sample size of 29 (using SPSS power analysis). Statistical analysis of the obtained data was performed using SPSS^®^25 (IBM Corporation, Armonk, NY, USA. Released, 2017).

## 3. Results

### 3.1. Changes from Baseline to Treatment Week 5

The ISI showed a significant improvement (decrease) from baseline to week 5 of treatment, with a very large effect size. Also, the sleep quality index showed a significant improvement (increase), but not the restorative sleep index (Table 2). Significant improvements (decreases) were seen for most of the ISI items. For the Karolinska Sleep Diary items, “ease of falling asleep” and “slept through” showed significant improvements (increases). The number of “Awakenings” showed a significant improvement (decrease), and the KSS at bedtime showed a significantly higher level of sleepiness (increase).

For the polysomnographic variables, sleep efficiency increased significantly, and WASO, TIB, number of awakenings, sleep latency, and stage N2 % decreased significantly (Table 3). The eta^2^ value was considerable for most significant variables.

### 3.2. Correlations between Changes in Subjective Ratings and PSG Changes from the Baseline to Treatment Week 5

Table 4 shows that improvements in the (subjectively rated) sleep quality index were significantly correlated with a decreased time in bed, decreased wake after sleep onset, and increased sleep efficiency. Improvements in the restorative sleep index were found to be significantly correlated with an increased sleep efficiency, decreased number of awakenings, and decreased N3 (in minutes as well as percentage). Changes in the ISI did not show any significant correlations with changes in the PSG variables.

Among the individual items of the Karolinska Sleep Diary, changes (improvements) in subjectively rated “sleep quality” correlated significantly with increased PSG-derived sleep efficiency, decreased TIB, and decreased WASO (Table 5). Changes in “calm sleep” showed the same pattern. Increased “ease falling asleep” was significantly correlated with decreased PSG-derived sleep latency. An increase in being “well rested” correlated significantly with a decreased PSG-derived number of awakenings/h, with decreased N3 (in minutes as well as percentage) and increased N2 (in minutes). Changes in the items “slept through”, “sufficient sleep”, and “ease awakening” did not show any significant correlations with changes in the PSG variables.

Among the individual items of the ISI, only the item “interference with daily functioning” correlated significantly with some of the PSG variables. Here, a decrease in TIB and N2 and an increase in N3 were associated with increased interference with daily functioning (Table 6). No other significant correlations were seen.

### 3.3. Selected Correlations of Possible Interest

Some further correlations between variables may be of interest when interpreting the results. Changes in the PSG variables were significantly correlated with the pre-treatment PSG values (r > 0.60, *p* ≤ 0.001) for all variables, except for the sleep stages (in minutes or percentages of TST). Thus, low initial sleep continuity (sleep efficiency, etc.) was associated with a larger increase in sleep continuity across the treatment period. The ISI correlated r = 0.10 (ns) between baseline and treatment week 5, the sleep quality index correlated r = 0.08 (ns) between the same points, and the restorative sleep index correlated r = 0.01 (ns). The change in the ISI from baseline to treatment week 5 was not significantly correlated with changes in the sleep quality index (r = 0.17, ns), nor with the restorative sleep index (r = −0.05 ns), and the ISI did not show any statistically significant correlation with the PSG variables at pre-treatment.

In the Appendix A, we present the correlations between the subjective and objective variables cross-sectionally at baseline and at treatment week 5. It is notable that the number of significant correlations were substantially more frequent for the baseline analysis. Particularly, PSG-based sleep efficiency and REM sleep showed significant correlations with subjective variables.

## 4. Discussion

The main purpose of this study was to investigate the association between changes in subjective and objective (PSG) sleep across a period of treatment with reduced TIB as a method to treat patients with insomnia. The change in the sleep quality index correlated with several PSG variables representing sleep continuity, as did changes in several of the items of the sleep quality index. Significant correlations were also seen for the restorative sleep index, but with somewhat different PSG variables. The change in the ISI total score was not significantly correlated with changes in the PSG variables, but change in one of its items, interference in daily functioning was correlated. The ISI and most of its items improved significantly from baseline to treatment week 5, as did, to a lesser extent, the PSG variables that reflected sleep continuity (sleep efficiency, WASO, number of awakenings/h, etc.), as well as the diary-rated sleep quality index and some of its items.

The significant correlation between the PSG sleep continuity variables and the sleep quality index was expected from previous studies [5,7,8,9,10,11,12,13]. The more detailed focus on the individual items of the sleep quality index indicates that improvements in sleep continuity measures, like PSG sleep efficiency and WASO, are linked to ratings of “sleep quality” and “calm sleep”, as in many of the previous studies. These two items have a “global character” in comparison with more specific ratings like “difficulties falling asleep” or “slept through”, suggesting that the best representation of PSG changes may be found in global items. Scores on the items “sleep quality” and “calm sleep” also increased significantly with decreasing TIB, which is the variable that was experimentally manipulated (reduced) in the treatment intervention. This suggests a causal link, and indicates that patients experience that their sleep quality increases and their sleep becomes calmer (or less restless) when their time in bed is shortened, which is a clinically very relevant finding. It is, however, worth remembering that this reflects changes over a five-week period, during which the vast majority of insomnia patients had finished sleep restriction or sleep compression (i.e., reduced TIB) and probably stabilized their sleep habits to a lower TIB. If the observations had been when the patients initiated the reduction in TIB before new sleep habits had been stabilized, then this correlation might have not been found or possibly been in the opposite direction.

It should be emphasized that the lack of a control group prevents us from solely attributing changes to the restriction of time in bed. Other factors may have contributed as well. This should not, however, affect the conclusion that changes in the subjective sleep ratings across the five weeks were well associated with changes in the PSG variables.

It is also of interest that changes in PSG-derived sleep latency correlated with changes in subjectively rated ease of falling asleep, as well as with (increased) sleepiness at bedtime. While logical, since it is the intent of the intervention, it suggests that the reduced time in bed may be linked to a shorter sleep latency through increased sleepiness/sleep pressure [19], but these results require confirmation in future studies.

The lack of association between change in sleep quality ratings and sleep architecture variables (sleep stages, in percent or minutes) was expected since such links have rarely been seen in previous work.

The significant association between changes in restorative sleep (index or individual items) and changes in the PSG variables was expected from our previous two studies [12,13], but ratings of restoration is rather uncommon in previous work, and there is little other data to compare with. An observation of particular interest was the decrease in N3 % and N3 minutes with increased restoration, including the item “well rested” (also seen in our longitudinal study [13]). One may speculate that sleep inertia, associated with awakenings from N3 sleep [20], may be involved. Thus, restitution should perhaps not be reported immediately upon awakening, but rather later during the day.

The fact that participants seemed to have an objective basis for judging the quality of a particular sleep should be of value in clinical work, but the long-term clinical usefulness of this knowledge remains to be demonstrated. This would require long-term studies using subjective and objective measures in a clinical setting.

Our finding of essentially a lack of association between changes in the ISI and PSG variables could have been expected from similar results in cross-sectional studies [2,3,21], but the present study hypothesized that an association might be brought out through the approach with within-subject change. This was not the case for the total index or its individual items (with one exception, see below). We cannot determine the reason for this lack of association, but one may argue that one night of recorded sleep may not be representative of the experience of sleep integrated across a week (or longer). The ISI also includes higher level and more abstract concepts, such as satisfaction with one’s sleep or worries about one’s sleep, than specific symptoms of disturbed sleep. However, its items on difficulties falling asleep, repeated awakening with difficulties going back to sleep, and too early awakenings are lower-level aspects on sleep, and changes in these variables did not show any significant associations with PSG changes. An interesting possibility is also that other factors than objective sleep continuity may play a role in the ratings of a week (or month) of sleep. This may involve other types of PSG variables that have yet to be identified. It may also be the case that subjective variables that reflect the participants’ experience of the total setting of sleep during an extended period may be important. The question of what constitutes subjective sleep quality across longer periods, and how the concept of sleep quality relates to the concept of insomnia disorder, clearly needs addressing in future studies.

The significant PSG/ISI item correlation (Table 5) that was found suggests that the reduction in TIB interfered with daily activities. This seems to reflect a discomfort with the reduction in TIB during all or parts of the treatment period, perhaps related to the increase in sleepiness discussed above and an increase in N3. The latter may be associated with a feeling of sleep inertia, as discussed above. The link with decreased N2 may simply reflect the reduction in TIB. A steep reduction in TIB, such as in sleep restriction therapy, can also come with a reduction in total sleep time/TST [14], which may be related to negative side effects during the day [22]. It is possible that the short treatment period of this PSG study (5 weeks) will include one or several adverse events during the daytime, especially when initiating the reduction in time in bed, and that these events are still rather fresh in memory and affects the specific ISI item about daytime interference. It is interesting that this finding, to some extent, contradicts the above finding, where decreased TIB correlates with subjectively better sleep quality and calm sleep, but these seemingly conflicting results might be due to the short study period, which is enough to stabilize new and better sleep habits, while patients still might have recently experienced negative daytime effects after nights with reduced sleep.

The negative correlation between baseline and treatment week 5 for the sleep continuity variables (PSG) indicates that those with the most disturbed sleep improved the most. This seems logical since the potential for improvement would be highest among those individuals. In contrast, the ISI, the sleep quality index, and the restorative sleep index showed nonsignificant correlations between pre- and post-treatment points of measurement. This indicates that individuals differed greatly in their subjective responses to sleep reduction.

It should be emphasized that the improvement in sleep measures from baseline to treatment week 5 cannot, in this particular study, be interpreted as an effect of the reduced time in bed, since no control group/condition was employed to establish such an effect. However, a causal effect of reduced time in bed (i.e., sleep restriction therapy or sleep compression therapy) has been previously demonstrated in several studies [23] and, whatever the reason, the improvement across time was pronounced, particularly for the ISI.

The present work has several limitations. One is the modest size of the sample, which makes generalizability difficult and prevents the analysis of subgroups. It is also possible that the two treatment approaches may have increased error variance, despite the two treatment groups being similar at pre- and post-treatment measurement. While the focus of this study was on the direct link between changes in subjective and objective measures, it would have been interesting to have included possible modifiers, like measures of psychological wellbeing/health. The supervised treatment period was relatively short, and it is possible that an extended period would provide further information of interest. In our analyses, we used both composite and single item scores, being aware of the psychometric advantages of the former. Still, we had a strong focus on single items, since they are likely to differ in their association with other variables, and thus may provide more nuanced information. One might also question the use of parametric analyses with subjective ratings across a five-step Likert scale. In the present case, however, we worked with changes in the range from −3 to +4 depending on the variable, and in most cases, their distributions were relatively normal. The number of correlations was large, and the number of significant correlations was low. This leaves some doubt with respect to our conclusions of these associations. Still, most of the findings seem logical and, at least partially, in line with the main hypothesis. Established correction techniques, like, for example, Bonferroni corrections, will increase the risk of type II errors. It should be emphasized that these results were obtained in a clinical group. The obtained results may have been different had a non-clinical group been investigated.

## 5. Conclusions

In conclusion, the results suggest that changes in improved sleep quality, as measured using a sleep diary, are linked to PSG changes (improvements) in sleep continuity measures for the same sleep. Associations for changes in restitution from sleep show a similar pattern, with an exception for N3, which seems to be associated with a poorer restitution. Changes in the ISI showed almost no association with changes in polysomnography for a particular recorded night of sleep, possibly due to one recorded sleep not being representative of a number of rated sleeps, or to the ISI items being conceptually different with their focus on insomnia disorder.

## Figures and Tables

**Figure 1 brainsci-13-01426-f001:**
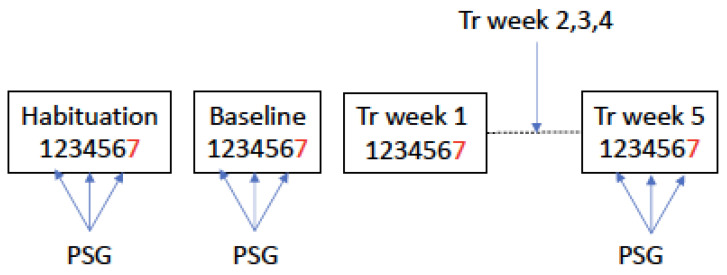
Flow chart of the study. Baseline and 5 treatment weeks. PSG was recorded on either Tuesday, Wednesday, or Thursday in the baseline week, or weeks 2, 5, and 10 (weeks 2 and 10 are not reported here). The ISI was administered at the end of the baseline week and each treatment week (“7”). The numbers 1–7 represent days of the baseline and treatment weeks. “Tr” = treatment. Note that “weeks” do not represent calendar weeks, but the weeks from the start of treatment.

**Table 1 brainsci-13-01426-t001:** Background data. N = 34.

Variable	Treatment, %/Mean ± SD
Age	44.4 ± 13.1 years
Women	75%
Married/cohabiting	86.1%
University education	63.9%
Employed/studying	97.2%
Hypnotics/sedatives (ever)	54.1%
Good economy	83.3%
ISI at screening	20.2 ± 3.7

ISI = insomnia severity index.

**Table 2 brainsci-13-01426-t002:** Changes observed between baseline and week 5 for subjective ratings.

Variable	Baseline	Week 5	F Ratio	Eta^2^
ISI: sum (0–24)	18.0 ± 0.67	11.1 ± 0.78	20.9 ***	0.72
Sleep quality index: mean (1–5)	3.16 ± 0.13	3.60 ± 0.13	5.6 *	0.17
Restorative sleep index: mean (1–5)	2.47 ± 0.12	2.67 ± 0.12	1.5	0.05
Karolinska Sleep Diary items (1–5, except KSS)				
Awakenings, number of	2.57 ± 0.34	1.75 ± 0.33	6.4 *	0.19
KSS at bedtime (1–9)	6.79 ± 0.25	7.48 ± 0.20	6.0 *	0.18
Ease falling asleep	3.41 ± 0.25	3.93 ± 0.19	5.4 *	0.17
Slept through	3.35 ± 0.30	3.93 ± 0.22	4.3 *	0.13
Sleep quality	2.82 ± 0.19	3.17 ± 0.19	1.8	0.06
Calm sleep	3.07 ± 0.17	3.35 ± 0.23	1.6	0.05
Sufficient sleep	2.48 ± 0.17	2.62 ± 0.19	0.4	0.01
Well rested	2.14 ± 0.12	2.38 ± 0.13	2.2	0.07
Ease awakening	2.79 ± 0.18	3.00 ± 0.17	0.9	0.01
KSS at rising (1–9)	6.51 ± 0.28	6.00 ± 0.25	2.1	0.07
ISI items (0–4)				
Difficulty staying asleep	2.45 ± 0.21	1.28 ± 0.15	32.7 ***	0.54
Dissatisfied with sleep	3.17 ± 0.14	2.14 ± 0.17	23.5 ***	0.46
Worried about sleep problems	2.62 ± 0.18	2.00 ± 0.19	8.5 **	0.23
Problem waking up too early	1.86 ± 0.22	1.31 ± 0.17	9.1 **	0.25
Difficulty falling asleep	1.52 ± 0.18	1.00 ± 0.19	6.5 *	0.19
Sleep problems interfere with daytime activities	2.31 ± 0.18	1.97 ± 0.20	3.4	0.11
Sleep problems noticeable by others	1.62 ± 0.18	1.45 ± 0.20	1.1	0.04

* = *p* < 0.05, ** = *p* < 0.01, and *** = *p* < 0.001. ISI = insomnia severity index. KSS = Karolinska sleepiness scale. Analysis of variance, F ratio, *p*-value, mean ± standard errors, and partial eta^2^. For the Karolinska Sleep Diary items, high values are positive (1–5), except for the number of awakenings, while the opposite is true for the ISI items (0–4, from positive to negative). N = 34.

**Table 3 brainsci-13-01426-t003:** Changes between baseline and week 5 for PSG variables.

Variable	Baseline	Week 5	F Ratio	Eta^2^
Sleep efficiency, %	83.3 ± 1.8	90.2 ± 1.2	18.7 ***	0.40
WASO, min	58 ± 8	31 ± 5	16.5 ***	0.37
TIB, min	452 ± 13	396 ± 9	14.6 ***	0.34
Sleep latency, min	16.1 ± 3.1	8.3 ± 1.2	7.0 *	0.21
N2, %	51.2 ± 1.9	47.6 ± 2.0	6.9 *	0.20
N2, min	191 ± 11	170 ± 10	5.2 *	0.16
Awakenings/h	4.7 ± 0.6	3.5 ± 0.35	4.6 *	0.14
REM, %	21.0 ± 1.0	23.1 ± 1.0	2.8	0.09
TST, min	368 ± 15	357 ± 10	0.5	0.02
N1, %	14.4 ± 1.9	14.1 ± 2.1	0.1	0.00
N3, %	13.4 ± 1.5	15.2 ± 1.3	1.7	0.06
Arousals/h	10.9 ± 1.3	10.2 ± 0.9	0.5	0.02
REM, min	79 ± 6	83 ± 4	0.4	0.01
N1, min	52 ± 6.0	50 ± 7	0.1	0.00
N3, min	49 ± 6	55 ± 5	1.3	0.05

* = *p* < 0.05 and *** = *p* < 0.001. TIB = time in bed, WASO = wake after sleep onset, TST = total sleep time, and min = minutes. F ratio, *p*-value, mean ± standard error, and partial eta^2^. N = 34.

**Table 4 brainsci-13-01426-t004:** Correlation between changes from baseline to treatment week 5 between subjective sleep and PSG variables. N = 34.

PSG Variable	Change in the ISI	Change in the SQI	Change in the RSI
TIB	−0.09	−0.47 **	−0.12
WASO	0.03	−0.33	−0.35
TST	−0.10	−0.06	0.25
Sleep latency	−0.11	−0.36 *	−0.12
Sleep efficiency	0.06	0.41 *	0.41 *
Number of awakenings/h	0.20	−0.06	−0.37 *
REM, %	0.05	0.19	0.30
N1, %	−0.06	−0.17	−0.05
N2, %	−0.15	−0.16	0.23
N3, %	0.15	0.11	−0.46 **
REM, min	−0.04	0.15	0.36 *
N1, min	−0.12	−0.23	0.01
N2, min	−0.15	−0.04	0.37
N3, min	0.21	0.07	−0.45 *

* = *p*< 0.05 and ** = *p* < 0.01. ISI = insomnia severity index, SQI = sleep quality index, and RSI = restorative sleep index. TIB = time in bed, WASO = wake after sleep onset, TST = total sleep time, and min = minutes.

**Table 5 brainsci-13-01426-t005:** Correlations between change (from baseline to treatment week 5) in Karolinska Sleep Diary items and change in PSG variables. N=34.

Change: PSG Variables	Change: Sleep Quality	Change: Calm Sleep	Change: Well Rested	Change: Slept Through	Change: Difficult Asleep	Change: Awakenings	Change: Enough Sleep	Change: Ease Awakening
TIB, min	−0.48 **	−0.48 **	0.13	−0.43 *	−0.18	0.24	−0.04	−0.33
WASO, min	−0.57 ***	−0.52 **	−0.24	−0.21	−0.08	0.19	0.09	−0.08
TST, min	−0.03	0.03	0.44 *	−0.11	−0.02	0.08	0.15	0.05
Sleep lat., min	−0.26	−0.16	0.03	0.06	−0.49 **	0.20	−0.04	−0.24
Sleep eff, %	−0.56 ***	0.49 **	0.32	0.19	0.20	−0.13	0.19	0.16
NAI	−0.19	−0.22	−0.40 *	−0.03	0.06	−0.08	−0.07	0.05
REM, %	0.29	0.18	0.28	−0.37 *	−0.11	0.08	0.36	−0.03
N1, %	−0.18	−0.21	0.10	−0.29	0.10	0.0	−0.20	−0.01
N2, %	−0.18	−0.12	0.17	−0.20	0.21	0.03	0.02	0.36
N3, %	0.06	0.13	−0.50 *	0.09	−0.17	0.08	−0.19	−0.32
REM, min	0.23	0.08	0.32	0.16	−0.07	0.12	0.27	0.07
N1, min	−0.23	−0.17	0.07	−0.24	0.02	0.04	−0.20	−0.07
N2, min	−0.04	−0.04	0.40 *	−0.15	0.14	0.08	0.20	0.15

* = *p* < 0.05, ** = *p* < 0.01, and *** = *p* < 0.001. TIB = time in bed, WASO = wake after sleep onset, TST = total sleep time, and min = minutes. NAI = number of awakenings index (per h).

**Table 6 brainsci-13-01426-t006:** Correlations for change from baseline to treatment week 5 between the ISI items and PSG variables. N = 33 (one outlier removed).

Change in the ISI							
Change in PSG Variable	Difficulty Falling Asleep	Difficulty Maintaining Sleep	Problems with Early Awakenings	Dissatisfied with Sleep	Sleep Problems Interfering	Sleep Problems Noticeable	Worried about Sleep Problems
TIB, min	0.07	0.15	0.02	−0.15	−0.36 *	0.09	−0.03
WASO, min	0.01	0.20	0.08	−0.07	−0.30	−0.12	−0.13
TST, min	0.01	0.08	0.08	−0.17	−0.26	−0.07	−0.06
Sleep latency, min	−0.19	0.22	0.12	−0.16	−0.30	−0.11	−0.12
Sleep efficiency, %	0.02	−0.21	−0.04	0.07	0.26	0.12	0.09
Number aw/h	0.13	0.11	−0.02	0.10	0.20	0.11	0.03
REM, %	−0.10	−0.05	0.07	0.01	0.22	0.15	−0.10
N1, %	−0.01	0.23	−0.11	0.15	−0.29	−0.06	−0.03
N2, %	0.19	−0.15	−0.07	0.01	−0.30	−0.10	−0.30
N3, %	0.01	0.14	0.05	0.06	−0.13	−0.11	−0.13
REM, min	−0.10	0.05	−0.04	−0.15	0.04	0.09	0.06
N1, min	−0.02	0.15	−0.13	−0.19	−0.32	−0.06	−0.04
N2, min	0.13	−0.04	0.13	−0.11	−0.38 *	−0.13	−0.22
N3, min	−0.05	0.00	0.17	0.18	0.35 *	−0.01	0.21

* = *p* < 0.05, TIB = time in bed, WASO = wake after sleep onset, TST = total sleep time, and min = minutes.

## Data Availability

Sharing of the data is not possible for ethical reasons (no consent on sharing in ethical application).

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
