# Peer review of "The Polysomnographical Meaning of Changed Sleep Quality—A Study of Treatment with Reduced Time in Bed"

_brainsci, 2023, doi:10.3390/brainsci13101426_

Round 1

Reviewer 1 Report

Review for „The polysomnographical meaning of changed sleep quality. A study of treatment with reduced time in bed”

The paper leverages an intervention designed to improve sleep to investigate how well changes in objective sleep are reflected in changes subjective sleep ratings, indirectly investigating how well subjective ratings reflect objective sleep quality.

The study is about an important topic and its design (correlation of change scores in response to an intervention) is novel. However, at N=34 it is underpowered. In my review I’ll attempt to suggest improvements which can increase the fidelity of findings.

Specific comments

1.       Introduction/Discussion: I believe that authors should cite a few more highly relevant studies about their topic. For example, Unruh et al 2008 (10.1111/j.1532-5415.2008.01755.x) used a very large sample to report close to zero correlations between PSG metrics and habitual sleep. Shirota et al (2022, 10.1007/s41105-022-00437-x) presented an interesting study in which natural within-person, day-to-day changes in objective and subjective sleep metrics were correlated. This design is similar to what the authors implement here and together with Unruh et al 2008 underscore the importance of using within-person differences in immediate, not habitual sleep as the metric objective metrics are supposed to reflect. McCarter et al 2022 10.1016/j.smrv.2022.101657 recently published an extensive literature review which the authors should definitely know about. They also conclude that subjective and objective sleep only correlates weakly at best, but I think between-subject designs are not ideal for investigating this because they are biased by trait-level confounders. For example, it is possible that older people, women, or those with a more neurotic or pessimistic personality either only report or only experience different sleep patterns, which biases correlations downward.

2.       What was the correlation between objective and subjective sleep metrics (not the changes!) on the two measurement occasions? This could directly contrast the within-individual and between-individual approaches and may demonstrate the inferiority of the latter.

3.       A full correlation matrix – the correlation of all objective and subjective metrics (maybe omitting items), both within- and across-measurements – should be presented in the Supplement. It would be especially interesting to see how well the different subjective sleep metrics correlated and how stable these and objective sleep metrics were across time.

4.       I would advise caution about correlations with individual items. Basic psychometric theory dictates that items have much lower reliability than composite scores as in the proportion of true to error variance is better (trait variance is common for each item so it adds up but error variance is unique so it cancels out) so any correlation with them is expected to be lower. That said, the pattern of correlations makes sense so it is possible that due to their very transparent content these items have high reliability. If the authors choose to present cross-occasion correlations at the item level, they may investigate how much lower these correlations are for items than composite scores.

5.       I think the authors can be more aggressive in delivering the point that using between-participant designs is suboptimal for studying the relationship between subjective and objective sleep, and may point to biases in response tendencies (for instance, reports of poor sleep partially reflecting personality as opposed to actual sleep) as the cause. Their findings, even with such a small sample, demonstrate quite strong correlations, and resonate well with the recent findings by Shirota et al. I’m unaware of another within-person study but my own recent findings, soon to be published as a preprint, also show that up to 40% of within-person variability in subjective morning diary sleep ratings can be accounted for by objectively measured sleep efficiency, TST and WASO alone, while between-person correlations are much weaker.

6.       A minor point: line 331 reads “asleepg”.

Author Response

R1

The paper leverages an intervention designed to improve sleep to investigate how well changes in objective sleep are reflected in changes subjective sleep ratings, indirectly investigating how well subjective ratings reflect objective sleep quality.

The study is about an important topic and its design (correlation of change scores in response to an intervention) is novel. However, at N=34 it is underpowered. In my review I’ll attempt to suggest improvements which can increase the fidelity of findings.

Specific comments

  1. Introduction/Discussion: I believe that authors should cite a few more highly relevant studies about their topic. For example, Unruh et al 2008 (10.1111/j.1532-5415.2008.01755.x) used a very large sample to report close to zero correlations between PSG metrics and habitual sleep. Shirota et al (2022, 10.1007/s41105-022-00437-x) presented an interesting study in which natural within-person, day-to-day changes in objective and subjective sleep metrics were correlated. This design is similar to what the authors implement here and together with Unruh et al 2008 underscore the importance of using within-person differences in immediate, not habitual sleep as the metric objective metrics are supposed to reflect. McCarter et al 2022 10.1016/j.smrv.2022.101657 recently published an extensive literature review which the authors should definitely know about. They also conclude that subjective and objective sleep only correlates weakly at best, but I think between-subject designs are not ideal for investigating this because they are biased by trait-level confounders. For example, it is possible that older people, women, or those with a more neurotic or pessimistic personality either only report or only experience different sleep patterns, which biases correlations downward.

R: Thank you for the suggestions! We had considered Unruh et al but refrained from using since it only included sleep duration and sleep latency. We have added McCarter, but could not find Shirota et al. although we tried several approaches.

  1. What was the correlation between objective and subjective sleep metrics (not the changes!) on the two measurement occasions? This could directly contrast the within-individual and between-individual approaches and may demonstrate the inferiority of the latter.

R: We computed the correlations and entered the results in the supplementary. We felt that, even if interesting, it was out of the scope of the present study. We hope this is acceptable.

  1. A full correlation matrix – the correlation of all objective and subjective metrics (maybe omitting items), both within- and across-measurements – should be presented in the Supplement. It would be especially interesting to see how well the different subjective sleep metrics correlated and how stable these and objective sleep metrics were across time.

R: We understand this thinking, but the correlation matrix would be very large and we would prefer not to include it, even in the supplementary, if acceptable. If not, we will provide it, of course.

  1. I would advise caution about correlations with individual items. Basic psychometric theory dictates that items have much lower reliability than composite scores as in the proportion of true to error variance is better (trait variance is common for each item so it adds up but error variance is unique so it cancels out) so any correlation with them is expected to be lower. That said, the pattern of correlations makes sense so it is possible that due to their very transparent content these items have high reliability. If the authors choose to present cross-occasion correlations at the item level, they may investigate how much lower these correlations are for items than composite scores.

      R: We certainly agree that psychometric theory tells us that composite scores have a higher reliability than single item scores. However, composite scores may hide differences between items in their association with another variable, and we were very much interested in the associations of the individual items since we feel that they tell us more about sleep than composite scores (we also included the composite scores). We have added a sentence to limitations to acknowledge our awareness of the lower reliability of single items.

  1. I think the authors can be more aggressive in delivering the point that using between-participant designs is suboptimal for studying the relationship between subjective and objective sleep, and may point to biases in response tendencies (for instance, reports of poor sleep partially reflecting personality as opposed to actual sleep) as the cause. Their findings, even with such a small sample, demonstrate quite strong correlations, and resonate well with the recent findings by Shirota et al. I’m unaware of another within-person study but my own recent findings, soon to be published as a preprint, also show that up to 40% of within-person variability in subjective morning diary sleep ratings can be accounted for by objectively measured sleep efficiency, TST and WASO alone, while between-person correlations are much weaker. 

R: We appreciate this comment, but did not design the study to compare within – and between study subjective/objective correlations. We, therefore, feel uncomfortable going beyond the conclusion of within-person correlations alone. We hope this is acceptable.

  1. A minor point: line 331 reads “asleepg”.

R:corrected

Reviewer 2 Report

The research topic is interesting from a clinical and scientific point of view. However, the manuscript needs serious correction.

Major comments

Formulate the purpose of the study and add it to the abstract and at the end of the Introduction section.

I recommend adding a flowchart of the design of this study to the Materials and Methods section.

Add an explanation of the sample size calculation to the Materials and Methods section. What methodology did you use? Why is the sample size small (only 34 participants)?

The main problem of this study is the absence of a control group without treatment (without reduction of time in bed) and/or a comparable group with an alternative treatment method (for example, pharmacotherapy or non-drug therapy).

Minor comments

Line 25 - Replace "EEG" with "PSG".

Lines 30, 31 and further in the text - Use the MDPI template for references (please use the reference numbers in square brackets).

Line 109 - The name of the Table 1 needs correction. Also, add the name of the first column in this table.

Lines 214, 224 - Add a clear name to Table 2 and Table 3. Please move the explanations to the Note below these tables. Add the name of the first columns.

Line 235 - The name of the Table needs revision.

Line 338 - Replace "polysomnography" with "PSG".

The style of the English language needs to be improved.

Author Response

R2

The research topic is interesting from a clinical and scientific point of view. However, the manuscript needs serious correction.

Major comments

1.Formulate the purpose of the study and add it to the abstract and at the end of the Introduction section.

R: We added: “In contrast, the purpose of this study was to investigate the association between chan ge in subjective and objective sleep variables

2.I recommend adding a flowchart of the design of this study to the Materials and Methods section.

R: We have added a flowchart as figure 1 and hope it is acceptable.

  1. Add an explanation of the sample size calculation to the Materials and Methods section. What methodology did you use? Why is the sample size small (only 34 participants)?

R: The sample size was determined by the number of patients in the Stockholm region that accepted the offer of a sleep recording in their treatment process.

4.The main problem of this study is the absence of a control group without treatment (without reduction of time in bed) and/or a comparable group with an alternative treatment method (for example, pharmacotherapy or non-drug therapy). 

R: This is correct, of course, but the purpose was to investigate the association between change in subjective/objective sleep variables within an insomnia group. It would have been good to have included a non-clinical group, but that was outside the scope of the present study. With respect to the evaluation of change with treatment, we emphasize that we cannot comment on any treatment effect since we lack a control group.

Minor comments

5.Line 25 - Replace "EEG" with "PSG".

R: Done

6.Lines 30, 31 and further in the text - Use the MDPI template for references (please use the reference numbers in square brackets).

R: Done

11.Line 109 - The name of the Table 1 needs correction. Also, add the name of the first column in this table.

R: Done

12.Lines 214, 224 - Add a clear name to Table 2 and Table 3. Please move the explanations to the Note below these tables. Add the name of the first columns.

R:Done

13.Line 235 - The name of the Table needs revision.

R: Done

14.Line 338 - Replace "polysomnography" with "PSG".

R: Done

Comments on the Quality of English Language

  1. The style of the English language needs to be improved.

R: We have consulted a native English speaker

Reviewer 3 Report

The article presents a study that explores the relationship between subjective sleep quality and objective measures of sleep obtained through polysomnography (PSG). The study focuses on patients with insomnia who underwent treatment involving reduced time in bed, aiming to investigate how changes in PSG variables correspond to changes in subjective sleep ratings.

It is important to consider the limitations of the study and I suggest article revisions.

Firstly, the sample size is relatively small, which may limit the generalizability of the findings. A larger and more diverse sample would provide a more robust basis for drawing conclusions. Please pay more attention to it in Limitations.

Secondly, the study design lacks a control group, making it difficult to establish a causal relationship between reduced time in bed and improvements in sleep quality. Without a control group, it is challenging to determine whether the observed changes are solely attributable to the treatment or influenced by other factors. Please pay more attention to it in Limitations.

Thirdly, the study relied on self-reported measures for subjective sleep quality and insomnia symptoms. Self-report measures can be subject to biases, and the study did not investigate objective measures of psychological well-being or other potential confounding variables that could affect sleep quality. Please pay more attention to it in Limitations.

Furthermore, the study only examined changes over a relatively short 5-week treatment period. Sleep habits and perceptions of sleep quality may continue to evolve over longer periods, and it would be valuable to explore how these changes persist or fluctuate over time. Please pay more attention to it in Limitations.

Finally, while the study did find correlations between changes in subjective and objective sleep measures, the clinical significance of these correlations is not entirely clear. Future research should aim to establish not only statistical associations but also the practical relevance of these findings for individuals with sleep problems. Please pay more attention to it in Limitations.

In conclusion, while the study offers insights into the relationship between reduced time in bed and improvements in sleep quality, it is limited by its small sample size, lack of a control group, and reliance on self-reported measures. The findings suggest a need for further research with larger samples and longer-term follow-up to better understand the complex interplay between subjective and objective sleep measures in the context of insomnia treatment. 

Also, authors should check the layout and formatting of the article.

***

Your reviewer.

Inconsistent use of tense: The text occasionally switches between past and present tense, which can be confusing. Ensure that the tense is consistent throughout.
Lack of clarity: Some sentences are quite complex and may benefit from rephrasing to enhance clarity. The choice of synonims is misleading in some places.

Author Response

R3

The article presents a study that explores the relationship between subjective sleep quality and objective measures of sleep obtained through polysomnography (PSG). The study focuses on patients with insomnia who underwent treatment involving reduced time in bed, aiming to investigate how changes in PSG variables correspond to changes in subjective sleep ratings.

It is important to consider the limitations of the study and I suggest article revisions.

1.Firstly, the sample size is relatively small, which may limit the generalizability of the findings. A larger and more diverse sample would provide a more robust basis for drawing conclusions. Please pay more attention to it in Limitations.

R: Now added: One is the modest size of the sample, which makes generalizability difficult and prevents analysis of subgroups.

  1. Secondly, the study design lacks a control group, making it difficult to establish a causal relationship between reduced time in bed and improvements in sleep quality. Without a control group, it is challenging to determine whether the observed changes are solely attributable to the treatment or influenced by other factors. Please pay more attention to it in Limitations.

R:True- We have now added the following text line 303: “, It should be emphasized that the lack of a control group prevents us from attributing changes solely to the restriction of time in bed. Other factors may have contributed as well. This should not, however, affect the conclusion that change in subjective sleep ratings across the 5 weeks were well associated with changes in PSG variables.

3,Thirdly, the study relied on self-reported measures for subjective sleep quality and insomnia symptoms. Self-report measures can be subject to biases, and the study did not investigate objective measures of psychological well-being or other potential confounding variables that could affect sleep quality. Please pay more attention to it in Limitations.

R: Now added: ” While the focus of the study was on the direct link between change in subjective and objective measures, it would have been interesting to have included possible modifiers, like measures of psychological wellbeing/health”.

  1. Furthermore, the study only examined changes over a relatively short 5-week treatment period. Sleep habits and perceptions of sleep quality may continue to evolve over longer periods, and it would be valuable to explore how these changes persist or fluctuate over time. Please pay more attention to it in Limitations.

R: Yes, we agree. We added this text to limitations:” The supervised treatment period was relatively short, and it is possible that an extended period would provide further information of interest.”

  1. Finally, while the study did find correlations between changes in subjective and objective sleep measures, the clinical significance of these correlations is not entirely clear. Future research should aim to establish not only statistical associations but also the practical relevance of these findings for individuals with sleep problems. Please pay more attention to it in Limitations.

R: Yes, now added as a separate pargraph in discussion  (line 325): “The fact participants seemed to have an objective basis for judging the quality of a particular sleep should be of value in clinical work, but the long-term clinical usefulness of this knowledge remains to be demonstrated. This would require long-term studies using subjective and objective measures in a clinical setting..”

  1. In conclusion, while the study offers insights into the relationship between reduced time in bed and improvements in sleep quality, it is limited by its small sample size, lack of a control group, and reliance on self-reported measures. The findings suggest a need for further research with larger samples and longer-term follow-up to better understand the complex interplay between subjective and objective sleep measures in the context of insomnia treatment. 

R: Agree

  1. Also, authors should check the layout and formatting of the article

R: We have tried to improve layout and formatting and hope it is acceptable.

Round 2

Reviewer 2 Report

The size of the samples should be calculated based on the current recommendations for the planning of scientific research, and not on the basis of how many people agreed to participate (according to the researchers).

For example, online calculators can be used to calculate the minimum number of samples needed to meet the required statistical constraints (https://www.calculator.net/sample-size-calculator.html , http://www.raosoft.com/samplesize.html , etc.).

This study has serious limitations, which should be separated into a separate section, since the clinical and scientific significance of the study is statistically poorly substantiated.

In addition, the manuscript needs a technical revision: the abstract is still not structured; the name of Figure 1 needs revision (it contains duplicate information); the authors did not use the MDPI recommended template for tables and references.

Author Response

1.The size of the samples should be calculated based on the current recommendations for the planning of scientific research, and not on the basis of how many people agreed to participate (according to the researchers). For example, online calculators can be used to calculate the minimum number of samples needed to meet the required statistical constraints (https://www.calculator.net/sample-size-calculator.html , http://www.raosoft.com/samplesize.html , etc.).

R: (as now added to statistics in Methods): There are no available data on correlations between changes for the present type of variables, but using cross-sectional correlations as an estimate, a power of 0.8. a significance level of p<.05, for an assumed  correlation of r=0.50, we would need a sample size of 29 (using SPSS power analysis).

2.This study has serious limitations, which should be separated into a separate section, since the clinical and scientific significance of the study is statistically poorly substantiated.                                    

R: limitations have been described in the paragraph of limitations. Should it be done in any other way?

3.In addition, the manuscript needs a technical revision: the abstract is still not structured (Done); the name of Figure 1 needs revision (it contains duplicate information) (Done); the authors did not use the MDPI recommended template for tables and references.

R:We can’t find any template for tables and figures, only the rather vague instructions on tables and figures (in the instruction document). We have done our best to follow those instructions. We have also adapted the references according to the instructions.

Reviewer 3 Report

I would like to thank the authors for considering all my comments.

Author Response

I would like to thank the authors for considering all my comments

R: Thank you!